# Associations between Serum Kallistatin Levels and Markers of Glucose Homeostasis, Inflammation, and Lipoprotein Metabolism in Patients with Type 2 Diabetes and Nondiabetic Obesity

**DOI:** 10.3390/ijms25116264

**Published:** 2024-06-06

**Authors:** Hajnalka Lőrincz, Sára Csiha, Balázs Ratku, Sándor Somodi, Ferenc Sztanek, György Paragh, Mariann Harangi

**Affiliations:** 1Division of Metabolism, Department of Internal Medicine, Faculty of Medicine, University of Debrecen, H-4032 Debrecen, Hungary; 2Doctoral School of Health Sciences, University of Debrecen, H-4032 Debrecen, Hungary; 3Department of Emergency Medicine, Faculty of Medicine, University of Debrecen, H-4032 Debrecen, Hungary; 4Institute of Health Studies, Faculty of Health Sciences, University of Debrecen, H-4032 Debrecen, Hungary; 5ELKH-UD Vascular Pathophysiology Research Group 11003, University of Debrecen, H-4032 Debrecen, Hungary

**Keywords:** kallistatin, triglyceride, lipid subfractions, diabetes, obesity, betatrophin

## Abstract

Kallistatin is an endogenous serine proteinase inhibitor with various functions, including antioxidative, anti-inflammatory, and anti-atherosclerotic properties. To date, associations between kallistatin and lipoprotein subfractions are poorly investigated. In this study, we enrolled 62 obese patients with type 2 diabetes (T2D), 106 nondiabetic obese (NDO) subjects matched in gender, age, and body mass index, as well as 49 gender- and age-matched healthy, normal-weight controls. Serum kallistatin levels were measured with ELISA, and lipoprotein subfractions were analyzed using Lipoprint^®^ (Quantimetrix Corp., Redondo Beach, CA, USA) gel electrophoresis. Kallistatin concentrations were significantly higher in T2D patients compared to NDO and control groups. We found significant positive correlations between very-low-density lipoprotein (VLDL), small high-density lipoprotein (HDL) subfractions, glucose, hemoglobin A_1c_ (HbA_1c_), betatrophin, and kallistatin, while negative correlations were detected between mean low-density lipoprotein (LDL) size, large and intermediate HDL subfractions, and kallistatin in the whole study population. The best predictor of kallistatin was HbA_1c_ in T2D patients, high-sensitivity C-reactive protein (hsCRP) and betatrophin in NDO patients, and hsCRP in controls. Our results indicate that kallistatin expression might be induced by persistent hyperglycemia in T2D, while in nondiabetic subjects, its production might be associated with systemic inflammation. The correlation of kallistatin with lipid subfractions may suggest its putative role in atherogenesis.

## 1. Introduction

Kallistatin belongs to the human tissue kallikrein inhibitor protein family. The human kallistatin gene is mapped to chromosome 14q31-32.1 [1] and encodes a 40 kDa protein expressed in the human liver, stomach, pancreas, kidneys, aorta, testes, prostate, arteries, atria, ventricles, lungs, renal proximal tubular cells, and a colonic carcinoma cell line [2]. Kallistatin signals through several receptors, including integrin β3, lipoprotein receptor-related protein 6, nucleolin, and Krüppel-like factor 4. Considering the complexity of its receptors, it is not surprising that kallistatin participates in the regulation of multiple signaling pathways and cell functions, such as angiogenesis, inflammation, and tumor growth [3].

Previous studies have demonstrated that circulating kallistatin levels are significantly elevated in patients with type 1 diabetes mellitus or prediabetes and positively correlated with fasting blood glucose and carotid intima-media thickness [4]. In these patients, higher kallistatin levels may represent a compensatory response to hyperglycemia [5]. Furthermore, kallistatin levels were significantly higher in type 2 diabetic (T2D) patients with vascular complications compared to nondiabetic control subjects and T2D patients without complications. Kallistatin levels in diabetic patients showed correlations with various clinical parameters related to vascular health, including hemoglobin A_1c_ (HbA_1c_), and large and small artery elasticity [6]. Moreover, serum kallistatin levels were significantly increased in T2D patients with diabetic nephropathy compared to T2D patients without nephropathy and healthy controls. In the whole T2D patient population, there was a positive correlation between the serum kallistatin level and the serum very-low-density lipoprotein cholesterol (VLDL) [7]. Serum levels of kallistatin were higher in obese individuals with prediabetes but were similar in subjects with and without insulin resistance, which indicates that the main factor for increased kallistatin levels may be the hyperglycemia and not the insulin sensitivity state [5].

Both T2D and obesity are associated with disturbed lipid metabolism characterized by hypertriglyceridemia and decreased high-density lipoprotein cholesterol (HDL-C) levels. Indeed, the abovementioned atherogenic dyslipidemia, including elevated levels of triglyceride-rich lipoproteins, such as VLDL and intermediate-density lipoprotein (IDL), increased number of small-dense low-density lipoprotein (LDL) particles, and decreased HDL-C level, is a major driving force of atherogenesis [8,9]. In healthy adolescents, the circulating kallistatin concentration was negatively correlated with total cholesterol and LDL-C levels but showed a positive correlation with HDL-C levels [10]. Furthermore, an inverse relationship between the plasma kallistatin concentration and total cholesterol levels was also observed in heart transplant recipients [11]. Based on these data, kallistatin may have a role in maintaining a favorable lipid profile [3]. However, the impact of kallistatin on lipoprotein metabolism, including HDL, still needs to be clarified.

Therefore, we aimed to determine the associations of kallistatin with lipoprotein subfractions and the markers of glucose metabolism in patients with T2D and nondiabetic obesity (NDO) and in healthy control subjects. We hypothesized that the serum kallistatin level is higher in diabetic patients and correlates with glycemic status and atherogenic lipoproteins, including triglyceride-rich lipoproteins, and may also be associated with HDL subfractions.

## 2. Results

Anthropometric data and the main laboratory parameters of the enrolled subjects are summarized in Table 1. Normal weight controls, NDO, and T2D patients were matched in gender and age. Body mass index (BMI) and waist circumference were markedly higher in the T2D and NDO groups than in the controls, while these variables were comparable in T2D and NDO patients. After Bonferroni correction, significantly higher triglyceride, glucose, HbA_1c_, insulin, hsCRP, gamma-glutamyl transpeptidase (γ-GTP), and alanine transaminase (ALT) levels were observed in the T2D group compared to the controls, whereas HDL-C levels were lower in the T2D group than the controls. Analyzing lipoprotein subfractions using Lipoprint^®^ gel electrophoresis, the percentages of large LDL, small-dense LDL, and small HDL were found to be significantly higher in both the NDO and T2D groups compared to the controls. Mean LDL size and the percentage of large HDL were significantly lower in T2D patients than in controls. Serum betatrophin concentrations were significantly higher in T2D patients than in the control and NDO groups, whereas concentrations were similar in the control and NDO groups (Table 1).

The main medications of the enrolled participants are summarized in Appendix A. The majority of T2D patients received metformin (72.8%), angiotensin-converting enzyme inhibitor/angiotensin II receptor blocker (ACEI/ARB) (45.2%), and statin therapy (41.9%), while one-third of the T2D patients were treated with insulin or glucagon-like peptide-1 receptor agonist. Due to hypertension, 38.7% and 20.8% of the NDO patients received ACEI/ARB and diuretic therapy, respectively.

Serum kallistatin levels were significantly lower in controls (7.09 ± 1.91 μg/mL) and NDO subjects (7.08 ± 1.96 μg/mL) compared to T2D patients (8.21 ± 1.70 μg/mL; *p* = 0.004 and *p* = 0.010, respectively; Figure 1).

Pearson’s correlations of kallistatin with the main anthropometric and routine laboratory parameters were performed in each study group and summarized in Table 2. In controls, kallistatin correlated negatively with hsCRP. In the NDO group, there was a positive correlation between betatrophin and kallistatin and a negative correlation between hsCRP and kallistatin. Moreover, kallistatin correlated positively with HbA_1C_ and triglyceride, while a negative correlation was found between HDL-C and kallistatin in T2D patients.

Pearson’s correlations of kallistatin with lipoprotein subfractions obtained by Lipoprint^®^ are shown in Table 3. Kallistatin showed significant positive correlations with the percentage of VLDL in overall, T2D, and NDO groups, while there was no correlation among controls (Table 3 and Appendix A). Also, similar tendencies were observed regarding the percentages of IDL and small-dense LDL in these groups (Table 3). The mean LDL size correlated negatively with kallistatin in overall, T2D, and NDO groups, while no correlation was found in controls (Table 3 and Appendix A). Regarding HDL subfractions, kallistatin showed significant correlations mostly with the intermediate and small-sized HDL subfractions in T2D patients (Table 3). There were negative correlations with the large-sized and positive correlations with small-sized HDL subfractions in NDO subjects. On the contrary, in the controls, kallistatin correlated mostly with intermediate-sized HDL subfractions (Table 3).

Serum kallistatin correlated negatively with the percentage of large and intermediate HDL subfractions in all subjects (Appendix A). There was a positive correlation between kallistatin and the percentage of small HDL in all subjects (Appendix A).

In addition, there were positive correlations between glucose, HBA_1c_, betatrophin, and kallistatin in all subjects (Figure 2).

We performed backward stepwise multiple regression analyses to determine significant predictor(s) of kallistatin in the control, NDO, and T2D groups (Table 4). Three different models were built according to the results of univariate regression analyses indicated in Table 2 and Table 3. Model 1 included hsCRP and intermediate HDL (%) in the dependent variable list in controls, and hsCRP was found to be a predictor of kallistatin (β* = −0.54; *p* = 0.001). In the NDO group, model 2 included betatrophin, hsCRP, VLDL (%), IDL (%), mean LDL size, large HDL (%), and small HDL (%) in the dependent variable list. As for the results of model 2, betatrophin (β* = 0.362; *p* = 0.007) and hsCRP (β* = −0.31; *p* = 0.001) were identified as predictors of kallistatin in NDO patients. Model 3 included HbA_1c_, triglyceride, HDL-C, VLDL (%), IDL (%), mean LDL size, large HDL (%), and small HDL (%) in the dependent variable list. HbA_1c_ was found to be a predictor of kallistatin in the T2D group (β* = 0.489; *p* = 0.018).

## 3. Discussion

This is the first clinical study demonstrating correlations of serum kallistatin levels with lipid fractions and subfractions in T2D and NDO patients and controls. We verified that the serum kallistatin level was higher in T2D patients compared to NDO patients and controls and correlated with glycemic status. Moreover, we found that the serum kallistatin level correlated with atherogenic lipoproteins, including triglyceride-rich lipoproteins. Furthermore, it was associated with HDL subfractions.

Atherogenic dyslipidemia—characterized by higher triglycerides, VLDL, large LDL, small-dense LDL levels, and a smaller mean LDL size, as well as shifting toward small-sized HDL subfractions in NDO and obese T2D patients compared to lean controls—has been described in previous studies, and these finding are in line with the literature data [12,13]. Only one previous study reported a positive correlation between the serum kallistatin level and triglyceride-rich lipoproteins, i.e., VLDL in T2D patients with diabetic nephropathy [7]. We also found significant positive correlations between serum kallistatin levels and percentages of VLDL, IDL, and small-sized LDL subfractions in T2D as well as in NDO patients. Furthermore, kallistatin correlated positively with triglycerides and negatively with the HDL-C levels in T2D patients. Although formerly negative correlations were detected between total cholesterol and LDL-C levels, and a positive correlation with HDL-C levels in healthy adolescents [10], we did not find significant correlations between these parameters in NDO patients and controls. Considering the well-established role of low LDL size in cardiovascular risk prediction [14], the negative correlations between mean LDL size and kallistatin levels in T2D and NDO patients may support the antiatherogenic nature of kallistatin. The clinical impact of strong correlations between serum kallistatin and the ratios of HDL subfractions needs further research, since due to the complex nature of HDL, the function might be superior to the structure of this lipoprotein [15].

The elevated serum kallistatin level and its positive correlation with HbA_1c_ in our T2D patient group corroborates the results of previous studies and confirms the key role of persistent hyperglycemia in the upregulation of kallistatin production [5,6]. Betatrophin, also known as angiopoietin-like protein 8 (ANGPTL8), is a protein that mainly plays a role in lipid metabolism [16]. Elevated serum betatrophin levels have been reported previously in T2D and obese patients [13,17,18,19]. Therefore, its significant positive correlation with kallistatin in NDO patients and in the whole study population is probably not surprising but a novel finding. However, this association was not detected in T2D patients and controls.

Previously, the serum kallistatin levels were investigated in some special nondiabetic obese populations, but not in NDO patients. Significantly, a higher kallistatin level was found in patients with nondiabetic-obesity-related chronic kidney disease [20]. The serum kallistatin concentration was significantly higher in obese and overweight patients after myocardial infarction compared to controls. Moreover, serum kallistatin concentrations were associated with the triglyceride glucose index and anthropometric parameters associated with overweight and obesity [21]. A significantly elevated circulating kallistatin level was reported in overweight women with polycystic ovary syndrome [22]. However, in the present study, we did not find significantly higher circulating kallistatin levels in NDO patients compared to controls.

Previously, kallistatin levels positively correlated with hsCRP, tumor necrosis factor-α (TNF-α), and carotid intima-media thickness in both polycystic ovary syndrome and control groups, but did not show correlations with fasting blood glucose, 2 h glucose values during an oral glucose tolerance test, or HbA_1c_ [22]. Indeed, administration of recombinant kallistatin markedly reduced inflammatory responses in various animal models of arthritis, myocardial ischemia, hypertension, and septic shock [23,24,25,26,27]. Although the exact mechanism has not been elucidated, kallistatin has been shown to inhibit vascular inflammation and apoptosis by preventing TNF-α- and high-mobility group box protein 1-mediated expression of several inflammatory genes [25,27] and to attenuate the inflammatory response via the nuclear factor kappa B signaling pathway in rheumatoid arthritis [28]. In obese patients, kallistatin inhibited lipopolysaccharide- and TNF-α-induced inflammation in human adipocytes via downregulating the expression and secretion of key inflammatory markers [29]. Based on these data, tissue kallikrein might be involved in the regulation of inflammatory responses. Our result in NDO patients and controls showed significant negative correlations between kallistatin and hsCRP levels in T2D and NDO, supporting its putative anti-inflammatory effects. Indeed, the backward stepwise multiple regression analyses showed that hsCRP is a significant predictor of kallistatin in NDO and control groups, which corroborated the intimate interplay between systemic inflammation and kallistatin expression. Furthermore, several previous studies demonstrated the anti-inflammatory effect of statins characterized by hsCRP reduction [30,31]. It must be noted that hsCRP and statins were not significant cofounders in the T2D group; therefore, these variables were not selected for the multiple regression analysis.

Some limitations of the study must be mentioned. Further inflammatory markers should be measured in these patient populations to better characterize the putative role of inflammation in the regulation of the circulating kallistatin level. Although the correlations between lipoprotein subfractions and circulating kallistatin levels were significant, the role of kallistatin in common regulatory pathways and lipoprotein metabolism cannot be fully elucidated due to the cross-sectional nature of this study. Also, it must be noted that the sample size differed between the three study groups, which might have affected the *p*-values. Still, the results support the antiatherogenic effect of kallistatin in diabetes and obesity. The correlations between kallistatin levels and lipoprotein subfractions in all study groups, the measurement of kallistatin levels in NDO patients, and its correlations with betatrophin levels are considered novel findings.

On one hand, the correlation of the circulating kallistatin level with glycemic status and atherogenic lipoproteins in T2D underlines the importance of intensive and permanent glucose and lipid control and emphasizes the role of the new classes of agents, such as the glucagon-like peptide-1 receptor agonists and the sodium-glucose cotransporter-2 inhibitors, as well as the use of statins to moderate the atherosclerotic process. On the other hand, the correlation between kallistatin and hsCRP in NDO patients and controls verified that low-grade inflammation is an important treatment target to lower the residual cardiovascular risk in these populations.

## 4. Materials and Methods

### 4.1. Enrolment of Study Subjects

A study design flowchart of the enrolled subjects is depicted in Appendix A. Forty-nine normal-weight, healthy volunteers (13 males and 36 females; mean age: 43.2 ± 9.1 years; BMI: 24.7 ± 2.8 kg/m^2^) with normal routine laboratory parameters were enrolled in the study. One hundred and six NDO subjects (23 males and 83 females; mean age: 44.3 ± 12.5 years; BMI: 42.6 ± 8.1 kg/m^2^) and sixty-two obese T2D patients were also enrolled (22 males and 40 females; mean age: 47.6 ± 7.7 years; BMI: 43.1 ± 9.1 kg/m^2^). To confirm the nondiabetic status in control and NDO groups, we performed a routine 75 g oral glucose tolerance test after fasting for 12 h. Obesity was defined as BMI ≥ 30 kg/m^2^. Patients with T2D were diagnosed according to the American Diabetes Association (ADA) 2022 guidelines, as follows: fasting plasma glucose ≥ 7.0 mm/L or 2 h plasma glucose ≥ 11.1 mmol/L or HbA_1c_ ≥ 6.5% (48 mmol/mol) [32]. Antidiabetic treatment (oral antidiabetic agents, insulin, or both) and lipid-lowering drugs were assessed. All participants came from the general internal medicine outpatient clinic or the obesity and diabetes outpatient clinics at the Department of Internal Medicine, Faculty of Medicine, University of Debrecen, Hungary. The study was performed according to the Helsinki Declaration and all subjects provided informed consent. Permission to carry out this study was granted by the Regional Ethics Committee of the University of Debrecen and the Medical Research Council (registration numbers: DE RKEB/IKEB 5513B-2020 and IV/7989-1/2020/EKU, respectively).

Subjects with active endocrine, liver, kidney, pulmonary, neurological, gastrointestinal, acute infective, autoimmune disease, or malignancies were excluded. Pregnant and lactating women, current smokers, and regular alcohol consumers were also excluded. Regular alcohol consumers were defined as patients who self-reported their average alcohol consumption in standard drinks as ~12 g alcohol per week over the preceding 12 months. In addition to anthropometric and laboratory data, we also recorded the main medications of all enrolled subjects. Patients with T2D were treated either with antidiabetics (mostly metformin and glucagon-like peptide-1 receptor agonists) or with insulin.

### 4.2. Determination of Routine Laboratory Parameters

The participants were referred to scheduled medical appointments in the morning after an overnight fast. Venous serum and plasma samples were collected, and after a half-hour rest, samples were separated at 3500 g, 15 min, at +4 °C. Routine laboratory parameters, including glucose, HbA_1c_, insulin, hsCRP, estimated glomerular filtration rate (eGFR), γ-GTP, AST, ALT, supersensitive thyroid-stimulating hormone (sTSH), total cholesterol, triglyceride, LDL-C, and HDL-C, were performed with a Cobas c600 autoanalyzer (Roche Ltd., Mannheim, Germany) at the Department of Laboratory Medicine, Faculty of Medicine, University of Debrecen, Hungary. Reagents were purchased from the same vendor and tests were performed according to the recommendations of the manufacturer. Samples were kept in a 200 μL aliquot at −70 °C for subsequent measurements.

### 4.3. Determination of Lipoprotein Subfractions

Lipoprotein subfractions were distributed using Lipoprint^®^ gel electrophoresis (Quantimetrix Corporation, Redondo Beach, CA, USA) and analyzed by Lipoware Image SXM v.1.82 Software (Quantimetrix Corp., Redondo Beach, CA, USA), as previously described [33].

Between the peaks of VLDL (Rf = 0) and HDL (Rf = 1), three midbands (C through A, mainly corresponding to IDL subfractions) and a maximum of seven LDL subfractions (LDL-1-7) can be distributed during the LDL subfraction test. The percentage of large LDL (large LDL%) was defined as the summed percentages of LDL-1 and LDL-2, whereas the percentage of small LDL (small-dense LDL%) was defined as the sum of LDL-3-LDL-7.

During the HDL subfraction test, ten HDL peaks could be distributed between the VLDL + LDL (Rf = 0) and albumin peaks (Rf = 1). HDL-1-3 belonged to the percentage of large HDL, HDL4-7 belonged to the percentage of intermediate HDL, and HDL-8-10 belonged to the percentage of small HDL subfractions.

### 4.4. Determination of Kallistatin

Serum kallistatin levels were measured by a commercially available enzyme-linked immunoassay (ELISA) kit (Human Serpin A4/Kallistatin DuoSet ELISA, Cat: DY1669, R&D Systems, Abingdon, UK). Sera were applied in 10,000-fold dilution in duplicate according to the instructions of the manufacturer. The assay range was 125.0–8000 pg/mL.

### 4.5. Determination of Betatrophin

Serum betatrophin levels were measured by a commercially available ELISA kit (Human betatrophin ELISA, Cat: RD191347200R, BioVendor, Brno, Czech Republic) according to the instructions of the manufacturer. Samples were applied in 5-fold dilution in duplicate. The limit of detection was 0.244 ng/mL. The intra- and inter-assay variation coefficients were 5.4–9.4% and 2.4–5.3%, respectively.

### 4.6. Statistical Analyses

Statistical analyses were performed using the Statistica 13.5.0.17 software (TIBCO Software Inc., Tulsa, OK, USA). Graphs were made using GraphPad Prism 6.01 (GraphPad Prism Software Inc., San Diego, CA, USA). Statistical power was also conducted with the SPH Analytics online calculator (SPH Analytics LTD., Alpharetta, GA, USA) to validate the differences in serum kallistatin levels in the three subgroups. Sample size was assessed for one-way ANOVA with the following serum kallistatin levels: 7.1 μg/mL (for controls, group 1), 7.1 μg/mL (for NDO, group 2), and 8.2 μg/mL (for T2D, group 3), respectively, with SD = 1.9 μg/mL. The alpha level was 0.05, with 0.8 (80%) of desired power. Required sample sizes for groups 1 to 3 were calculated as n_1_ = 32, n_2_ = 80, and n_3_ = 40, respectively. The actual statistical power was above 0.8 (0.97). Normality of continuous data was checked with the Kolmogorov–Smirnov test, and the results were expressed as mean ± SD or median (interquartile range) in the descriptive table. The relationship between the male/female ratio was calculated with the Chi-square test. For the comparison of control, NDO, and T2D groups, one-way ANOVA (with Tukey’s post-hoc test) or the Kruskal–Wallis H test was performed. Bonferroni-adjusted significance tests were applied to adjust *p*-values for multiple comparison. When the calculated *p*-values were less than the Bonferroni threshold (*p* = 0.0022), the difference was considered statistically significant.

Pearson’s correlation was used to investigate the relationship between selected variables. Non-normally distributed variables were transformed logarithmically before correlation analysis. Backward stepwise multiple regression analyses were performed to determine significant predictor(s) of kallistatin in the overall population and different subgroups. *p* ≤ 0.05 probability values were considered statistically significant.

## 5. Conclusions

Our results confirmed the key role of persistent hyperglycemia in the elevated kallistatin concentrations found in T2D patients. The compensatory response to hyperglycemia seemed to overwrite the effect of inflammatory milieu characterized by hsCRP in T2D, whereas inflammation might be the main driving force of compensatory kallistatin production in nondiabetic subjects. The observed correlations between kallistatin and lipid fractions and subfractions may be weak, but they may imply the potential role of kallistatin signaling in lipoprotein metabolism, which corroborates the potential role of kallistatin in atherogenesis. Although these observational data cannot prove the direct causal link between kallistatin and lipid homeostasis, they raised its possibility and may inspire further research in this field. The pathogenic and clinical significance of kallistatin needs further investigation.

## Figures and Tables

**Figure 1 ijms-25-06264-f001:**
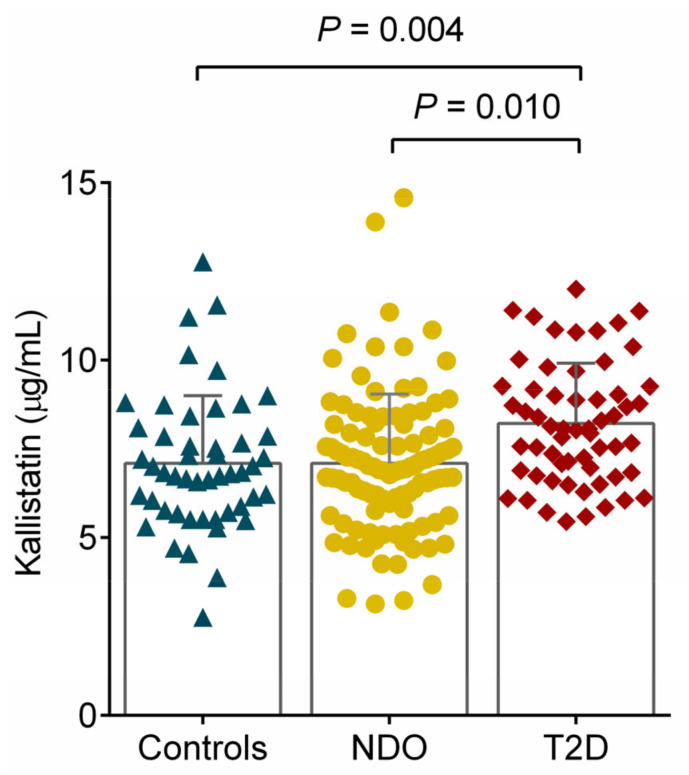
Mean concentrations of kallistatin in control (7.09 ± 1.91 μg/mL; marked with blue triangles), nondiabetic obese (NDO; 7.08 ± 1.96 μg/mL; yellow dots), and obese patients with type 2 diabetes (T2D; 8.21 ± 1.70 μg/mL; red squares). Differences were calculated using one-way ANOVA (*p* = 0.004 between controls vs. T2D and *p* = 0.010 between NDO vs. T2D, respectively). Columns represent means and whiskers represent standard deviations in the figure.

**Figure 2 ijms-25-06264-f002:**
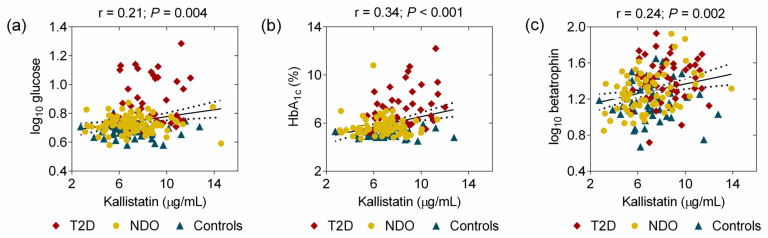
Correlations of kallistatin with (**a**) glucose, (**b**) HbA1c, and (**c**) betatrophin in all subjects. Red squares: obese patients with type 2 diabetes (T2D); yellow dots: nondiabetic obese (NDO) patients; blue triangles: controls. Solid lines represent the linear regression bands and dotted lines represent the 95 percent confidence intervals.

**Table 1 ijms-25-06264-t001:** Anthropometric and laboratory parameters of the study participants.

	Controls (*n* = 49)	NDO (*n* = 106)	T2D (*n* = 62)
Male/Female (n)	13/36	23/83	22/40
BMI (kg/m^2^)	24.7 ± 2.8	42.6 ± 8.1 *	43.1 ± 9.1 §
Waist circumference (cm)	85.2 ± 12.3	123.7 ± 17.4 *	128.3 ± 18.5 §
Age (years)	43.2 ± 9.1	44.3 ± 12.5	47.6 ± 7.7
**Lipid parameters**			
Total cholesterol (mmol/L)	5 ± 0.8	5 ± 0.8	5 ± 1.2
Triglyceride (mmol/L)	1.1 (0.9–1.5)	1.45 (1.1–1.9)	1.7 (1.2–2.7) §,#
LDL-C (mmol/L)	2.9 ± 0.5	3.2 ± 0.7	3 ± 0.9
HDL-C (mmol/L)	1.5 ± 0.4	1.3 ± 0.3	1.2 ± 0.3 §
**Lipoprotein subfractions**			
VLDL (%)	17.69 ± 3.2	19.9 ± 4.1	20.8 ± 5.2
IDL (C-A;%)	26.6 ± 6.3	25.3 ± 4	24.6 ± 3.9
Large LDL (1–2;%)	23.2 ± 6.1	28 ± 4.7 *	26.9 ± 5.4 §
Small-dense LDL (3–7;%)	0.6 (0–1.9)	1.15 (0–2.4)	1.65 (0–3.3) §
Mean LDL size (nm)	27.3 (27–27.4)	27.1 (26.9–27.3)	26.9 (26.9–27.3) §,#
Large HDL (1–3;%)	29 ± 8.7	22.9 ± 7 *	18.9 ± 6.5 §,#
Intermediate HDL (4–7;%)	50.3 ± 4.6	51.1 ± 3.7	48.8 ± 4.1
Small HDL (8–10;%)	20.7 ± 6.3	26 ± 6.8 *	32.2 ± 7.7 §,#
**Routine laboratory parameters**			
Glucose (mmol/L)	4.8 (4.5–5.1)	5.2 (4.9–5.8) *	6.4 (5.5–10.5) §,#
HbA_1C_ (%)	5.1 ± 0.3	5.7 ± 0.8	7.2 ± 1.7 §,#
Insulin (mU/L)	10.9 (6.6–12.9) (n = 16)	15 (11.2–21.6)	25.4 (14.1–31.5) (n = 16) §
hsCRP (mg/L)	1.3 (0.6–2.5)	8 (3.4–15.7) *	6.8 (3.1–13.7) §
eGFR (mL/1.73 m^2^)	90 (90–90)	90 (90–90)	90 (90–90)
sTSH (mU/L)	1.65 (1.18–2.11)	1.95 (1.46–2.67)	2.29 (1.34–15.2) (n = 33)
γ-GTP (U/L)	19 (16–28)	28.5 (19–44) *	35 (25–53) §
AST (U/L)	19 (17–24)	20 (17–27)	25 (17–30)
ALT (U/L)	17.5 (13–25)	26 (18–35) *	28 (21–44) §
Betatrophin (ng/mL)	15.35 (11.72–23.88)	18.22 (13.14–26.12)	26.85 (19.00–37.10) §,#

Abbreviations: ALT, alanine transaminase; AST, aspartate aminotransferase; BMI, body mass index; eGFR, estimated glomerular filtration rate; HbA_1C_, hemoglobin A1C; HDL-C, high-density lipoprotein cholesterol; hsCRP, high-sensitivity C-reactive protein; IDL, intermediate-density lipoprotein; LDL-C, low-density lipoprotein cholesterol; NDO, nondiabetic obese patients; sTSH, supersensitive thyroid-stimulating hormone; T2D, patients with type 2 diabetes mellitus; γ-GTP, gamma-glutamyl transpeptidase; VLDL, very-low-density lipoprotein. Notes: Data are presented as mean ± SD or median (interquartile ranges). Categorical variables were analyzed using the Chi-square test and continuous variables were analyzed using one-way ANOVA or the Kruskal–Wallis H test. * Indicates *p* < 0.0022 between controls vs. NDO; § indicates *p* < 0.0022 between controls vs. T2D; # indicates *p* < 0.0022 between NDO vs. T2D.

**Table 2 ijms-25-06264-t002:** Pearson’s correlations of kallistatin with anthropometric and routine laboratory parameters in the enrolled study groups.

	Controls (*n* = 49)		NDO (*n* = 106)		T2D (*n* = 62)	
Variable	r	*p*	r	*p*	r	*p*
**Antropometric parameters**						
Age (years)	−0.21	0.139	0.06	0.552	0.059	0.659
BMI (kg/m^2^)	−0.15	0.345	−0.07	0.455	−0.18	0.184
Waist circumference (cm)	0.14	0.417	−0.04	0.738	0.02	0.898
**Markers of glucose metabolism**						
Glucose (mmol/L)	−0.07	0.653	0.02	0.869	0.22	0.150
HbA_1C_ (%)	−0.01	0.978	−0.02	0.872	**0.40**	**0.009**
Insulin (mU/L)	0.30	0.260	0.13	0.209	−0.11	0.697
Betatrophin (ng/mL)	−0.03	0.841	**0.35**	**0.003**	0.07	0.611
**Markers of inflammation and liver enzymes**						
hsCRP (mg/L)	**−0.54**	**0.002**	**−0.23**	**0.030**	0.02	0.932
γ-GTP (U/L)	−0.08	0.642	−0.04	0.737	0.20	0.203
AST (U/L)	0.18	0.218	−0.02	0.883	0.16	0.305
ALT (U/L)	−0.01	0.971	0.01	0.904	0.23	0.126
**Lipid parameters**						
Total cholesterol (mmol/L)	0.15	0.304	0.13	0.201	0.19	0.196
Triglyceride (mmol/L)	−0.30	0.090	0.10	0.332	**0.46**	**0.015**
LDL-C (mmol/L)	0.11	0.460	0.10	0.333	0.06	0.706
HDL-C (mmol/L)	0.18	0.212	0.02	0.831	**−0.31**	**0.036**

Abbreviations: ALT, alanine transaminase; AST, aspartate aminotransferase; BMI, body mass index; HbA_1C_, hemoglobin A1C; HDL, high-density lipoprotein; hsCRP, high-sensitivity C-reactive protein; LDL, low-density lipoprotein; NDO, nondiabetic obese patients; T2D, patients with type 2 diabetes; γ-GTP, gamma-glutamyl transpeptidase. Notes: Non-normally distributed data were transformed logarithmically before Pearson’s correlation analysis. Significant data (*p* < 0.05) are bolded.

**Table 3 ijms-25-06264-t003:** Pearson’s correlations of kallistatin with lipoprotein subfractions obtained by Lipoprint^®^ in the enrolled study groups.

	Controls (*n* = 49)		NDO (*n* = 106)		T2D (*n* = 62)	
Variable	r	*p*	r	*p*	r	*p*
**LDL subfraction test**						
VLDL (%)	0.02	0.896	**0.40**	**<0.001**	**0.37**	**0.004**
IDL (C-A;%)	−0.13	0.378	**−0.25**	**0.010**	**−0.27**	**0.040**
Large LDL (1–2;%)	−0.04	0.802	−0.13	0.190	−0.18	0.170
Small LDL (3–7;%)	0.17	0.230	**0.26**	**0.007**	**0.30**	**0.022**
Mean LDL size (nm)	−0.10	0.482	**−0.28**	**0.004**	**−0.37**	**0.004**
**HDL subfraction test**						
HDL-1 (%)	0.01	0.990	**−0.28**	**0.004**	0.09	0.494
HDL-2 (%)	0.02	0.978	**−0.26**	**0.008**	−0.17	0.219
HDL-3 (%)	0.04	0.783	−0.08	0.439	**−0.37**	**0.005**
HDL-4 (%)	0.02	0.876	−0.05	0.644	**−0.34**	**0.009**
HDL-5 (%)	**−0.29**	**0.043**	0.08	0.444	**−0.28**	**0.034**
HDL-6 (%)	**−0.30**	**0.034**	0.02	0.857	−0.18	0.183
HDL-7 (%)	−0.06	0.671	−0.01	0.983	0.03	0.854
HDL-8 (%)	0.04	0.789	**0.34**	**<0.001**	**0.34**	**0.010**
HDL-9 (%)	0.09	0.539	**0.31**	**0.001**	**0.42**	**0.001**
HDL-10 (%)	**0.30**	**0.036**	0.12	0.226	**0.32**	**0.018**
Large HDL (1–3;%)	0.03	0.861	**−0.24**	**0.013**	−0.23	0.074
Intermediate HDL (4–7;%)	**−0.32**	**0.026**	0.02	0.835	**−0.32**	**0.014**
Small HDL (8–10;%)	0.20	0.170	**0.24**	**0.013**	**0.36**	**0.005**

Abbreviations: HDL, high-density lipoprotein; IDL, intermediate-density lipoprotein; LDL, low-density lipoprotein; NDO, nondiabetic obese patients; T2D, patients with type 2 diabetes; VLDL, very-low-density lipoprotein. Notes: Non-normally distributed data were transformed logarithmically before Pearson’s correlation analysis. Significant data (*p* < 0.05) are bolded.

**Table 4 ijms-25-06264-t004:** Determination of independent predictor(s) of kallistatin as a dependent variable using backward stepwise multiple linear regression analysis in controls, nondiabetic obese subjects (NDO), and patients with type 2 diabetes (T2D).

	Controls	NDO		T2D
	Model 1	Model 2		Model 3
Predictor	hsCRP	Betatrophin	hsCRP	HbA_1c_
β*	−0.54	0.362	−0.31	0.489
*p*-value	0.001	0.007	0.001	0.018

Model 1 included hsCRP and intermediate HDL (%) in the dependent variable list. Model 2 included hsCRP, betatrophin, VLDL (%), IDL (%), mean LDL size, large HDL (%), and small HDL (%) in the dependent variable list. Model 3 included HbA_1c_, triglyceride, HDL-C, VLDL (%), IDL (%), mean LDL size, large HDL (%), and small HDL (%) in the dependent variable list.

## Data Availability

All data generated or analyzed during this study are included in this published article. All data generated or analyzed during the current study are available from the corresponding author upon reasonable request.

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
