# Peer review of "Associations between Serum Kallistatin Levels and Markers of Glucose Homeostasis, Inflammation, and Lipoprotein Metabolism in Patients with Type 2 Diabetes and Nondiabetic Obesity"

_ijms, 2024, doi:10.3390/ijms25116264_

Round 1

Reviewer 1 Report

Comments and Suggestions for Authors

The authors examined associations between serum levels of kallistatin and serum levels of lipoprotein subfractions as well as various laboratory and two clinical parameters in healthy controls, nondiabetic obese patients (NDO) and patients with T2D diabetes.

The authors found several significant correlations. Using multivariable liner regression analyses the authors identified hsCRP, betatropin and HbA1c as independent predictors of kallistatin in controls, NDO, and T2D, respectively.

Major issues:

May major concern is a lack of novelty.

Majority of the data are confirmatory.

The authors failed to clearly highlight what is known and what is missing and therefore important to be addressed.

The authors also failed to highlight major findings- according to their hypothesis and the title of the manuscript one would expect first lipoproteins followed by the markers of glucose metabolism and finally CRP.

1.     Poor organization of the manuscript (please see below), inappropriate statistics (correction for multiple testing is missing; please see below), overinterpretation (the majority of the obtained correlations are weak, only two are moderate - almost all correlation coefficients are lower that 0.4)  and poor presentation of the data (not systematically described results, figures without clear logic for selecting data for presentation as a figure or supplemental table).

2.     The authors propose in the Introduction section to examine associations between kallistatin and lipoproteins as well as markers of glucose metabolism. The title also highlights lipoprotein. However, in the presented data lipoproteins do not play a major role; the associations were described only marginally. Therefore, results presented in the Table S2 should be presented in the main body of the manuscript as Table 2 with 2 parts.

First part: Correlations with markers of glucose metabolism

Second part: Correlations with Lipoprotein parameters obtained by Lipoprint

The Table should contain only 3 subgroups without overall; N of subjects in each group should be indicated in each table column as in Table 1.

P value should be corrected for multiple testing; considering 22 variables and 3 subgroups p<0.0022 should be considered significant.

Significant correlations between kallistatin and lipoproteins should be adjusted (partial correlations) for: 1. age, BMI; 2. age, BMI, hsCRP; 3. Age, BMI, HbA1c; 4. Age, BMI, betatropin.

The results presented in Table 2 should be concisely described in the text of the results section. In the original manuscript the authors show some correlations in figures and some in both figures and Table S2, without properly describing obtained data.

The authors should be aware of the fact that sample size differs between the groups and that this affects p-values.

Correlations between kallistatin and other clinical and route laboratory parameters should be shown in a separate Table.

Fig. 1 should be omitted

Figs 2-4 should be replaced with Table 2 as suggested above.

Weak correlations between kallistatin and lipoproteins, out of which only few will remain significant after correction for multiple testing + results of multivariable logistic regression showing that none of the lipoproteins is an independent predictor are results that the authors need to present, describe and discuss exactly as they are, without overinterpretation.  

Minors:

Lane 216: Previously, …..were indeed kallistatin levels positively correlated with hsCRP in that study; the authors found opposite in the present manuscript.

English should be improved in the whole manuscript.

Please provide the size (nm) of the presented HDL subfractions.

Comments on the Quality of English Language

Editing needed.

Reviewer 2 Report

Comments and Suggestions for Authors

Journal: IJMS       Manuscript ID: ijms-3011682

Authors: Hajnalka LÅ‘rincz et al.

Title: "Potential role of kallistatin in lipoprotein metabolism of patients with type 2 diabetes and nondiabetic obesity"

The authors of the present study explored the association between kallistatin, an endogenous serine proteinase inhibitor known for its potential antioxidative, anti-inflammatory, and antiatherosclerotic properties, among individuals with obesity and type 2 diabetes (T2D) or without (NDO), as well as healthy, normal-weight controls. According to the study findings, significantly higher levels of kallistatin were observed in T2D patients compared to both the NDO and control groups. Additionally, positive correlations were identified between kallistatin and certain lipoprotein subfractions, glucose, HbA1c, and betatrophin, while negative correlations were observed with other lipoprotein subfractions. The study hypothesis is intriguing; however, the following points merit consideration.

Comments:

1.     It would be important if the authors included in model 3 of the regression analysis hsCRP, antidiabetic medications, and statins (model 4).

2.     Please add a flow chart.

3.     Please clarify the inclusion criteria.

4.     Please provide a clear definition of "regular alcohol consumers," ensuring transparency and reproducibility.

5.    Did the authors conduct a sample size calculation?

Round 2

Reviewer 1 Report

Comments and Suggestions for Authors

The authors significantly improved the manuscript.

Minor issues

1. Title suggestions:

Version 1:

Serum kallistatin levels and their correlations associations with markers of

glucose homeostasis, inflammation and lipoprotein metabolism in patients with type 2 diabetes and nondiabetic obesity

Version 2:

Associations between serum kallistatin levels and  markers of

glucose homeostasis, inflammation and lipoprotein metabolism in patients with type 2 diabetes and nondiabetic obesity

2. Please label Table 2A as Table 2 and Table 2B as Table 3 as well as the current Table 3 as Table 4.

Sorry, for misunderstanding; the reviewer’s initial idea was to present the correlations data in one single Table with 2 parts: part 1 correlations with anthropometric and routine parameters and part 2 with lipoproteins.

Presently, it looks fine in 2 separate tables, but as mentioned above, these separate Tables should be Table 2 and Table 3.

Comments on the Quality of English Language

There are still some grammar and syntax errors.

Reviewer 2 Report

Comments and Suggestions for Authors

Journal: IJMS    Manuscript ID: ijms-3011682 (Revised version)

Authors: Hajnalka LÅ‘rincz et al.

Title: "Potential role of kallistatin in lipoprotein metabolism of patients with type 2 diabetes and nondiabetic obesity"

The authors of the present article have responded to my comments and suggestions and made the necessary changes to the paper, improving it further. It would be helpful if the following points could also be clarified in the revised paper:

-        Regarding my previous comment #1 and the authors' response 'In the iterative process of variable selection, covariates are removed from the model if they are non-significant and not a confounder,' did the authors consider the possibility that hsCRP and statins may act as potential confounders given their association with inflammatory processes and the potential involvement of kallistatin in inflammation?

-        The authors should add the relevant reference after their statement: 'Patients with T2D were diagnosed according to ADA guidelines.' In addition, they should clarify if they followed exactly the ADA criteria (e.g., including two measurements for T2D definition, etc.).

-        Regarding the sample size calculation, please provide further information and the necessary details in terms of reproducibility (e.g., the standard deviation, the desired power, alpha level, the statistical test) and the estimated sample size for each group.

These changes would further improve the clarity and comprehensiveness of the study.
